# The Role of Methionine Restriction in Gastric Cancer: A Summary of Mechanisms and a Discussion on Tumor Heterogeneity

**DOI:** 10.3390/biom14020161

**Published:** 2024-01-30

**Authors:** Yonghui Zou, Yiwu Yuan, Qi Zhou, Zhenqi Yue, Jiang Liu, Luojun Fan, Hesong Xu, Lin Xin

**Affiliations:** Department of General Surgery, The Second Affiliated Hospital of Nanchang University, No. 1 Minde Road, Donghu District, Nanchang 330006, China; 403007230040@email.ncu.edu.cn (Y.Z.); 363007210045@email.ncu.edu.cn (Y.Y.); zhouqihy@email.ncu.edu.cn (Q.Z.); 360014230112@email.ncu.edu.cn (Z.Y.); 350014230002@email.ncu.edu.cn (J.L.); 413007230003@email.ncu.edu.cn (L.F.); 413007230208@email.ncu.edu.cn (H.X.)

**Keywords:** gastric cancer, methionine, helicobacter pylori, programmed cell death, immune response, gastric cancer stem cells, chemotherapy, heterogeneity

## Abstract

Gastric cancer is ranked as the fifth most prevalent cancer globally and has long been a topic of passionate discussion among numerous individuals. However, the incidence of gastric cancer in society has not decreased, but instead has shown a gradual increase in recent years. For more than a decade, the treatment effect of gastric cancer has not been significantly improved. This is attributed to the heterogeneity of cancer, which makes popular targeted therapies ineffective. Methionine is an essential amino acid, and many studies have shown that it is involved in the development of gastric cancer. Our study aimed to review the literature on methionine and gastric cancer, describing its mechanism of action to show that tumor heterogeneity in gastric cancer does not hinder the effectiveness of methionine-restricted therapies. This research also aimed to provide insight into the inhibition of gastric cancer through metabolic reprogramming with methionine-restricted therapies, thereby demonstrating their potential as adjuvant treatments for gastric cancer.

## 1. Introduction

In China, gastric cancer ranks second in terms of incidence among all types of cancer, and it also holds the second-highest cancer-related mortality rate. Compared to most developed countries, China exhibits a higher mortality-to-incidence ratio (0.845) and a 5-year prevalence rate of 27.6 cases per 100,000 people [1]. Of even more significant concern is the escalating prevalence of early-onset gastric cancer (EOGC) in the young Chinese population [2].

There are many risk factors for gastric cancer [3], including *Helicobacter pylori* infection [4], smoking [5,6,7], dietary habits, abdominal obesity [8], alcoholism [9], and genes and genetics [9]. In China, with the improvement of sanitary conditions and the importance people attach to health, the incidence of gastric cancer was reduced by a decrease [10,11] in the prevalence of *Helicobacter pylori* infection [12], a decrease in preserved foods [13], etc. However, due to changes in living and working environments, as well as shifts in lifestyle, new factors contributing to the development of cancer have emerged [14]. Factors such as heightened work pressure, abdominal obesity, and unhealthy dietary habits (such as reliance on takeaways, processed meats, barbecue, fried food, and late-night snacks) have been identified as potential contributors to the increased incidence of gastric cancer [1,15,16,17,18]. Remarkably, these factors have shown a significant impact on the younger population (below 50 years old) in China. This particular form of gastric cancer is commonly referred to as early-onset gastric cancer [19]. It is believed to be linked to gastritis, the dysbiosis of the gastrointestinal microbiota, and the heightened utilization of antibiotics and acid suppressants [20]. According to anatomical location, gastric cancer can be categorized broadly as cardia gastric cancer, non-cardia gastric cancer, and gastroesophageal junction cancer. Likewise, based on the Lauren classification, gastric cancer can be classified into two main types: diffuse type and intestinal type [21]. According to the 2014 Cancer Genome Atlas Project, gastric cancer can be categorized into four subtypes: Epstein–Barr virus (EBV) type, microsatellite instability (MSI) type, chromosomal instability (CIN) type, and genome-stable type. These subtypes exhibit varying etiological and pathogenic mechanisms, which, in turn, have implications for treatment strategies and prognostic outcomes [22].

However, despite the increasing research being conducted on gastric cancer, the treatment outcomes of gastric cancer have not significantly improved from their previous states over a decade ago [23]. Early non-metastatic gastric cancer is primarily treated with endoscopic treatment or surgical resection [24]. For non-EGJ gastric cancer patients, the current standard treatment is D2 gastrectomy followed by adjuvant chemotherapy. Neoadjuvant therapy can be considered for patients with resectable advanced gastric cancer (C iii or above) [25]. For patients with advanced EGJ gastric cancer, neoadjuvant chemoradiotherapy is an option. Targeted therapies approved for the treatment of gastric cancer include trastuzumab as the first-line treatment for HER2-positive patients, ramucirumab as the second-line treatment for anti-angiogenesis, and nivolumab or pembrolizumab as the third-line treatment for anti-PD-1 [26]. Perioperative or adjuvant chemotherapy can improve the survival rate of patients with stage 1b cancer or higher [27]. Sequential chemotherapy, starting with platinum and fluoropyrimidine [28], is used for advanced gastric cancer. However, the median survival time remains less than 1 year. Additionally, many phase II and phase III clinical trials of molecular targeted drugs have failed. After analyzing the reasons, it was concluded that the heterogeneity of molecular characteristics among cancer cells in patients makes the efficacy of molecular targeted drugs highly unstable [29,30,31,32,33,34,35,36]. This suggests that the use of molecular targeted drugs in the treatment of gastric cancer may not be an ideal pathway. It is therefore an urgent problem to find a new approach to treat gastric cancer that can mitigate the influence of heterogeneity.

Methionine (Met) is one of the essential amino acids in the human body that cannot be synthesized in vivo. Numerous studies have demonstrated [37] the dependence of gastric cancer cells on Met, and an environment lacking Met is conducive to the efficacy of chemotherapy for gastric cancer. Furthermore, the γMetase enzyme can inhibit the proliferation of gastric cancer cells by cleaving methionine [38]. This suggests that Met may stimulate gastric cancer cell proliferation. Additionally, Met can undergo direct conversion to S-adenosyl methionine (SAM) through adenylyltransferase, which serves as a crucial direct methyl donor in the body. SAM plays a significant role in methylation processes, contributing methyl groups to over 50 substances within the body [39]. Therefore, Met can influence the progression of gastric cancer through various pathways. On the one hand, the conversion of Met to SAM can reverse proto-oncogene hypomethylation and subsequently decrease their expression, thereby inhibiting the proliferation of gastric cancer cells [40,41,42,43]. It has also been suggested that SAM is associated with ferric death in gastric cancer [44].

On the other hand, the overexpression of Metase significantly promotes apoptosis and autophagy, thereby inhibiting gastric cancer [40,45,46]. Additionally, Met can exert an impact on the development of gastric cancer by influencing the survival of *Helicobacter pylori* [47,48,49,50]. It has been shown that Met enhances the efficacy of the chemotherapeutic drugs 5-fluorouracil and Cisplatin [47,51,52,53]. The objective of this review is to analyze the role and mechanism of Met in the development of gastric cancer and to consider its potential implications in the treatment of gastric cancer.

## 2. The Relationship between Met and the Methionine Cycle

Met is an essential amino acid, and its metabolism is governed by the core cycle known as the methionine cycle. In this cycle, Met undergoes a conversion to SAM through the action of Met adenosyltransferase (MAT). SAM then serves as a methyl donor in methyltransferase reactions. A necessary byproduct of this process is S-adenosyl homocysteine (SAH) [54], which is hydrolyzed by the enzyme SAHH, resulting in the formation of homocysteine (Hcy). Hcy is eliminated from circulation either through conversion to homocysteine thiolactone or through the trans-sulfuration pathway, leading to the formation of cysteine. Alternatively, remethylation can occur through two pathways: one pathway utilizes methionine synthase with B12 as a cofactor and 5-methyltetrahydrofolate as the methyl donor, and the other involves betaine homocysteine methyltransferase with betaine as the methyl donor. Overall, the Met cycle connects and interacts with three vital metabolic pathways by supplying substrates (Figure 1).

In cancer cells, the deficiency of Met leads to the abnormal recognition of translation initiation sites, thereby inhibiting protein synthesis [55]. The significance of Met in cancer cell function has been extensively studied. Metabolites generated through the metabolic cycle play crucial roles in various intracellular processes, including polyamine synthesis, DNA synthesis, redox homeostasis, and methylation reactions [19]. SAM serves as a methyl donor for methyltransferases (MTases), enzymes that facilitate the transfer of methyl groups to various biomacromolecules, including DNA, RNA, proteins, and other metabolites that require methylation through the addition of methyl groups [56]. After losing the methyl group, SAM is converted to SAH. SAH has a significant impact on the catalytic ability of DNA methyltransferases (DNMTs) and histone methyltransferases (HMTs) [57]. SAH determines the activity of DNA and histone methyltransferases [54]. SAH undergoes hydrolysis to produce Hcy. Hcy has two potential fates; it can either enter the transsulfuration pathway or be methylated to form Met, thus connecting the Met cycle to the transsulfuration pathway. The entry of Hcy into the transsulfuration pathway leads to the production of glutathione (GSH), which plays a role in redox homeostasis. Reactive oxygen species (ROS) present in cancer cells can promote cancer cell proliferation through the PI3K pathway [58]. GSH functions as an antioxidant by reacting with ROS, scavenging them, and forming oxidized glutathione (GSSG) on its own [59].

## 3. Mechanisms of Met in the Pathophysiology of Gastric Cancer

The reliance on Met is evident in the gastric cancer cell line, which implies that leveraging this biochemical disparity between normal and malignant cells may have potential therapeutic implications [60].

Sufficient levels of Met are essential for human cells. In the event of inadequate exogenous methionine intake to sustain regular physiological processes, normal human cells can internally utilize the residual Hcy to synthesize Met [61]. Various studies have demonstrated that tumor cells experience a heightened demand for Met due to their rapid proliferation. However, tumor cells are incapable of utilizing Hcy to synthesize Met for several reasons, with the leading cause being the ineffectiveness of the Met synthase enzyme in cancer cells. Thus, tumor cells are more sensitive to the MR environment [60,62,63,64]. Nevertheless, our comprehension of the role of Met metabolism in cancer remains at an early stage due to its intricate nature. On a positive note, there has been a substantial increase in published studies in recent years, shedding more light on the impact of Met on gastric cancer.

### 3.1. Mechanisms of Met’s Role in Helicobacter pylori Survival and Infection

*Helicobacter pylori* (Hp) infection is a significant risk factor for gastric cancer and is classified as a Group 1 carcinogen by the International Agency for Research on Cancer (IARC) [4]. Following infection with Hp, the adjacent gastric epithelial cells undergo a complex inflammatory response. This environment triggers the release of various cytokines, reactive oxygen species (ROS), and nitric oxide (NO) by immune cells. These inflammatory mediators subsequently activate DNA methyltransferases, resulting in the hypermethylation of CpG islands and ultimately leading to the downregulation of associated gene expression [65]. Furthermore, gastric epithelial cells undergo epigenetic alterations following Hp infection, including the methylation of E-calmodulin, a process that may contribute to gastric cancer metastasis [66]. Epigenetic alterations significantly improve after Hp eradication, leading to a delay in the development of Hp-induced gastric cancer [67]. Hp faces multiple challenges to thrive in the stomach, such as the acidic environment and oxidative stress [68,69,70]. Hp possesses multiple mechanisms to counteract oxidative stress. One prominent mechanism involves the enzyme catalase, which facilitates the conversion of hydrogen peroxide and oxygen into less detrimental compounds. Prior research indicates that Met residues in peroxidases and ureases can serve a non-catalytic function to mitigate oxidative stress. The specific mechanism involves the binding of Met residues to oxidizing agents, resulting in the formation of methionine sulfoxide (Met-SO), which can subsequently be restored to methionine residues by methionine sulfoxide reductase (Msr). This establishes a Met-S/Met-SO cycle, effectively reducing oxidative damage in Hp [49,71].

Moreover, evidence shows that Hcy, a direct precursor of Met, synthesizes endogenous hydrogen sulfide (H_2_S) through a degradation pathway. Hcy is converted to H_2_S via three pathways: cystathionine β-synthase (CBS), cystathionine γ-lyase (CSE), and 3-mercaptopyruvate sulfotransferase (3MST) [72]. This metabolic process promotes vasodilation angiogenesis, inhibits leukocyte adhesion to vessel walls, and upregulates antioxidant molecules [73,74]. Some studies have suggested that H_2_S is a protective factor, mitigating chronic inflammation, including gastric mucosal atrophy induced by Hp [75]. H_2_S is a reducing agent that scavenges oxidizing molecules, including ROS and hydrogen peroxide. Additionally, H_2_S enhances wound healing by facilitating angiogenesis through the phosphatidylinositol 3-kinase/Akt signaling pathway [76,77,78]. Furthermore, H_2_S restricts neutrophil migration, inflammation, and oxidative burst [79] (Figure 2).

In a Met-rich environment, Met provides aid in Hp’s defense against oxidative stress, thus promoting Hp survival and activity. Conversely, Hcy-rich environments generate H_2_S, proving to be detrimental to Hp’s survival and activity. Theoretically, an MR strategy involving low Met and high Hcy could potentially delay gastric carcinogenesis by suppressing Hp survival and activity. The entire process does not directly involve gastric cancer cells; therefore, theoretically, the heterogeneity of gastric cancer cells would not affect the effectiveness of MR.

### 3.2. The Mechanism of Met in Programmed Cell Death of Gastric Cancer Cells

Programmed cell death, known as apoptosis, is a genetically determined, active, and orderly process through which cells die. This mechanism functions as a protective measure against stimuli from both internal and external environmental factors. There are four recognized types of programmed cell death: apoptosis, programmed necrosis, cellular pyroptosis, and ferroptosis. A study conducted in 2013 [80] observed that MR induces apoptosis and inhibits cell adhesion and migration in gastric cancer cells. The number of cancer cells cultured in the MR medium significantly reduced compared to the control group.

Furthermore, gastric cancer cells within the MR environment did not exhibit peritoneal spreading, and the same favorable outcome was observed in the in vivo experiments. Subsequent studies demonstrated that the MR medium fully demethylated the endogenous E-cadherin (CDH1) gene and enhanced CDH1 expression in the MKN45 and KATOIII cell lines. CDH1 is an intercellular adhesion protein. The promoter region of CDH1 is commonly hypermethylated in cases of diffuse gastric cancer, which is closely linked to the downregulation of CDH1 protein expression; moreover, clinical evidence has demonstrated that the downregulation of CDH1 protein expression leads to a more aggressive phenotype of gastric cancer and a worsened prognosis for patients with gastric cancer [81,82]. These studies demonstrated that MR induces apoptosis and suppresses the invasive metastasis of gastric cancer cells.

Recent studies [46] have indicated that the overexpression of the Met cleavage enzyme (Metase) significantly promotes apoptosis and autophagy in gastric cancer cells. Furthermore, the transfection of the SGC7901 and MKN45 cell lines with Metase substantially enhances apoptosis. Notably, the expression of Beclin1, Atg5, and Atg7 proteins and the autophagy marker LC3-I ratio were considerably elevated. Additionally, further investigations revealed that the expression of SNHG5 was increased, while miR-20a expression was reduced in Metase-transfected cancer cells, and there is clinical evidence that miR20a promotes cancer development [83]. Furthermore, the effect observed in cancer cells transfected with Metase was nullified upon adding si-SNHG. Notably, the silencing of SNHG5 using si-SNHG5 resulted in a decrease in miR-20a expression, thereby promoting the proliferation of gastric cancer cells [84,85]. Thus, it can be concluded that the depletion of Met in gastric cancer cells stimulates autophagy and apoptosis, thereby hindering the proliferation of gastric cancer cells.

It was discovered [45] that the expression of PI1K, the ratio of phosphorylated Akt (p-Akt) to total Akt (t-Akt), GLUT-2, and key glycolytic enzymes, such as HK2, PFKM, and LDHA, were downregulated in gastric cancer cell lines following treatment with recombinant methioninase (rMetase). Additionally, the anti-apoptotic protein Bcl-3 was downregulated, while the pro-apoptotic proteins Bax and cysteinyl asparagin-7901 were upregulated. Clinical evidence suggests that BCL-3 downregulation prolongs survival in gastric cancer patients [86]. Recent research has demonstrated that the gastric cancer cell viability and proliferation rate decreased, and the expression of long non-coding RNA (lncRNA) PVT1 was significantly reduced after culturing in a medium lacking Met. Furthermore, further investigations showed that the interaction between lncRNA PVT1 and DNMT3 impacted the DNA methylation level of the BNIP1 promoter. This interaction, along with the downregulation of lncRNA PVT1 and the upregulation of BNIP3 levels, inhibited the proliferation of gastric cancer cells, and clinical evidence demonstrates that BNIP3 upregulation favors the prognosis of cancer patients [87,88]. The deprivation of Met can upregulate the expression of BNIP1 by inhibiting the binding between lncRNA PVT3 and DNMT1, which, in turn, activates mitochondrial autophagy and ultimately inhibits the proliferation of gastric cancer cells [89].

Ferroptosis is a distinct form of non-apoptotic cell death regulation primarily triggered by excessive lipid peroxidation [90]. It is a recently identified programmed cell death mechanism mainly observed in tumor cells. Metabolic reprogramming has the potential to alter the sensitivity to iron-induced cell death. The metabolism of iron, lipids, and glutamine in cells is recognized as a crucial metabolic process that influences the vulnerability of cells to iron poisoning [90,91]. It is well known that cells are protected from iron oxidation by three principal antioxidant axes, i.e., the cystatin/GSH/GPX4 axis, FSP1/CoQ10 axis, and GCH1/BH4/DHFR axis [92]. Recent studies have shown that the inhibition of MR or the SAM-generating enzyme Met adenylyltransferase 2A (MAT2A) results in gastric cancer cells that are more susceptible to iron apoptosis. Ferroptosis inhibitors can eliminate this effect. Further studies revealed that MR-induced ferroptosis is not caused by cell cycle disruption or decreased cell viability, but it is a direct result of Met cycle disruption. The pharmacological inhibition of MAT2A resulted in a significant decrease in histone H3K4me3 trimethylation in the ACSL3 promoter region (acyl-coenzyme A synthetase long-chain family member 3), leading to the downregulation of ACSL3 expression [55]. ACSL3 plays a pivotal role as the central enzyme in the process of generating fatty acyl-coenzyme A esters. Moreover, incorporating these esters into membrane phospholipids provides a protective mechanism for cells against ferroptosis-induced mortality [93]. Thus, the activation of the MR receptor or the inhibition of the MATA2 protein promotes the occurrence of ferroptosis in gastric cancer cells. This promotion is achieved by suppressing the trimethylation of histone H3 at lysine 4 (H3K4me3), reducing the expression levels of ACSL3, and clinical evidence demonstrates that ACSL3 downregulation benefits the prognosis of cancer patients [44,94] (Figure 3).

When reviewing these studies, it becomes apparent that Met, being a critical methyl source, can modulate apoptosis, autophagy, and ferroptosis in gastric cancer cells by regulating genetic material and protein methylation. This effect is not limited to any specific molecule. Therefore, despite the heterogeneity observed among individual gastric cancer cells and even among gastric cancer cells within the same entity, it is theoretically improbable for MR to promote programmed cell death in gastric cancer cells.

### 3.3. The Mechanism of Met’s Role in the Immune Response to Gastric Cancer

Chronic inflammation, caused by the immune system, is widely recognized as a prominent factor in cancer development [95]. Consequently, cancer has been dubbed by certain scholars as “the persistent wound” [96]. Undoubtedly, the immune system plays a crucial role in the development and progression of cancer [97]. An illustrative example is the ability of *Helicobacter pylori* to induce chronic gastritis, thereby establishing H. pylori as a significant risk factor for the development of gastric cancer. Moreover, during the advanced stages of cancer, the immune system can stimulate cancer cell migration and augment the invasiveness of malignant cells [98]. The impact of the immune system on cancer extends to the tumor microenvironment. Cancer cells employ immune cells by inducing their cytokine release, leading to microenvironmental alterations that facilitate tumor angiogenesis and tissue remodeling [99,100]. Cancer cells are associated with the immune system due to both the heterogeneity between cancer cells and host cells and the rapid growth of cancer cells. When cancer cells grow at a pace that exceeds the energy supply in the bloodstream, necrosis occurs, resulting in the production of cellular debris and an abundance of pro-inflammatory factors. These factors activate the immune system, subsequently recruiting immune cells [101,102]. Moreover, it facilitates cancer development, entangled in a self-perpetuating loop. Nonetheless, the immune system also plays a role in suppressing cancer [103]. Upon exposure to tumor antigens, T cells become activated and differentiate into cytotoxic effector T cells, which are proficient in eliminating cancer cells. Consequently, T cells play a pivotal role in cancer immunotherapy [104].

Clinical and laboratory evidence suggests that cancer cells are capable of evading the immune system by assimilating significant quantities of Met from the environment [105,106,107]. It is not unexpected that MR therapy might have the potential to be utilized as an immunotherapy to restrain cancer progression [108].

It has been demonstrated that MR or inhibition of MATA2 reduce RIP1 expression by suppressing the trimethylation of lysine-4 on histone H3 (H3K4me3) and lysine-27 on histone H3 (H3K27me3) at the promoter of RIP1. As a result, this inhibition hinders the infiltration of monocytes/macrophages in gastric cancer [109]. Monocytes/macrophages are a significant component in the immune cell infiltration of gastric cancer [110]. Similarly, methionine exerts a substantial influence on T cell activity [111,112,113]. Upon antigen recognition, T cells undergo rapid proliferation and differentiation, requiring a substantial methyl supply from Met for efficient completion of DNA methylation and histone methylation [114,115]. Furthermore, studies have demonstrated that when T cells are stimulated in the absence of Met, there is a decrease in histone H3K4me3 modification. This reduction in histone methylation subsequently inhibits both T cell proliferation and differentiation [116]. In mouse Th17 cells, the activation of MR results in a significant reduction in the secretion of interleukin-17 (IL-17) and interferon-gamma (IFN-γ), thereby compromising the functionality of T cells. This has important implications in cancer tissues, as cancer cells heavily uptake the surrounding Met, thereby depriving T cells of an adequate supply of this essential nutrient. Consequently, the insufficiency of Met acquisition by T cells enables cancer cells to evade T cell-mediated elimination, thereby promoting their survival and progression [105]. In the earlier discussion, it was noted that normal cells can utilize homocysteine to synthesize methionine for their essential cellular functions, while cancer cells lack this capability. Consequently, by supplementing an adequate amount of homocysteine during MR in cancer cells, it is possible to hinder the proliferation of cancer cells and reactivate T cells to eliminate cancer cells. There have been previous studies showcasing the promotion of T cell proliferation and differentiation by Hcy. However, a research gap exists in terms of additional investigation [117,118].

Overall, the role of Met in gastric cancer immunization is crucial. MR not only inhibits the development of gastric cancer cells but also hampers the infiltration of monocytes/macrophages. Although the impact of Hcy supplementation on the effect of MR on gastric cancer cells remains unaffected, it alleviates the competitive inhibition of T cells caused by cancer cells and promotes T cell proliferation and differentiation. While gastric cancer cells display heterogeneity, immune cells in the body are our normal cells, and there is minimal variation in the same type of immune cells among different gastric cancer patients. Thus, the mechanisms influenced by Met are presumed to be generalizable.

### 3.4. Mechanisms Underlying the Impact of Met on Gastric Cancer Stem Cells

Gastric cancer stem cells (GCSCs) are a specific subtype of cancer cells characterized by their ability to self-renew and differentiate into other cancer cell types. It is widely acknowledged that the invasion, metastasis, and resistance to chemotherapy observed in gastric cancer can be attributed to the presence and activities of GCSCs. These cells play significant roles in the progression and treatment response of gastric cancer [119]. Due to the fast proliferation rate of GCSCs, the demand for Met is high. The proliferation rate of GCSCs was significantly reduced in mice fed with MR but exposed to Hcy. RAB37, a small GTPase protein, is responsible for cellular signaling and vesicular transport and can induce autophagy by binding to autophagy-related gene 5 (ATG5); in addition, clinical studies have shown that ATG5 downregulation promotes gastric cancer invasion, and the downregulation of RAB37 expression is associated with cancer metastasis and a poor prognosis [120,121,122]. Subsequent studies revealed that MR enhanced RAB37’s action through two pathways, promoting autophagy in GCSCs and resulting in a decrease in the proliferation rate of GCSCs. Firstly, MR decreased the methylation of RAB37, resulting in the upregulation of RAB37 expression. Additionally, regulating the miR-200b/PKCα axis indirectly increased the activity of RAB37. These actions led to increased autophagy and a decreased proliferation rate of GCSCs [40]. Additionally, Met can regulate the methylation of non-coding RNAs, thus affecting the proliferation of GCSCs. Various studies have demonstrated that miR-7, a non-coding RNA, functions as a tumor suppressor in different cancer types, including gastric cancer. Specifically, miR-7 suppresses the invasion and metastasis of gastric cancer through the inhibition of the epidermal growth factor receptor (EGFR) and nuclear factor-κB (NF-κB) signaling pathway [123,124]. There are also relevant clinical data that show that miR-7 inhibits gastric cancer proliferation [125]. Nevertheless, the expression of miR-7 in GCSCs was found to be suppressed due to an increase in DNA methylation of the miR-7 promoter in GCSCs. GCSCs cultured in an MR medium exhibited elevated miR-7-5p expression, which correlated with a reduced invasiveness of GCSCs. Subsequent studies revealed that MR treatment activates p53 signaling in GCSCs, selectively triggers apoptosis [126], and causes a decrease in the DNA methylation level of the miR-7-5p promoter region, thereby further promoting apoptosis in GCSCs [127] (Figure 4).

Multiple studies have demonstrated that MR can effectively inhibit the proliferation of GCSCs through various mechanisms. These findings indicate that MR has a profound impact on gastric cancer, and, importantly, this impact is not quickly diminished by the heterogeneity observed among gastric cancer cells.

### 3.5. The Role and Mechanism of Met in Gastric Cancer Chemotherapy

Despite the significant heterogeneity observed among cases of gastric cancer, there is minimal variation in the treatment approach for patients with advanced stages of the disease. Typically, for patients with limited advanced-stage cancer, the standard treatment involves resection followed by adjuvant chemotherapy [128]. Adjuvant chemotherapy commonly entails a regimen known as FLOT, which consists of docetaxel, oxaliplatin, 5-fluorouracil, and folinic acid. This combination of perioperative chemotherapeutic agents has become the standard of care in Western countries [129]. In cases of advanced diffuse gastric cancer, the option of neoadjuvant radiotherapy should be considered for patients who are deemed unsuitable for surgery [130]. In China, the majority of patients with gastric cancer who seek medical attention are already in advanced stages of the disease [131]. The effectiveness of chemotherapeutic drugs relies on tumor cells being in an active replication phase, which is essential for their action. However, in practice, only a small proportion of tumor cells are actively replicating, while the majority of cancer cells are in a quiescent state known as the G0/G1 phase. This quiescent state poses a significant challenge to the efficacy of chemotherapeutic drugs [132]. Moreover, cancer cells in the G0/G1 stage are more prone to migration compared to cells in other stages [133].

Met functions as an in vivo source of methyl donors. MR induces the entry of cancer cells, which are trapped in the G0/G1 phase, into the S/G2 phase, where they subsequently stagnate [134,135]. This is the period during which cellular DNA undergoes rapid replication, making it the prime time for cancer cells to be highly susceptible to chemotherapy. Thus, in theory, MR could augment the efficacy of chemotherapeutic drugs for treating cancer. A growing body of research has demonstrated that MR can synergistically interact with various chemotherapeutic agents; for instance, cancer cells in Met^−^/Hcy+ medium selectively arrested in the S/G2 phase and furthermore eliminated clonogenic cells and rendered cancer cells sensitive to cell-cycle-specific drugs [64], including 5-fluorouracil [136], gemcitabine [137], cisplatinum [52], Doxorubicin (DOX), Vincristine (VCR) [138], and so on. In addition to inducing the chemo-sensitive S/G2 phase, MR has the potential to augment the efficacy of chemotherapeutic agents through alternative pathways. Due to the challenge of achieving MR through regular feeding, researchers have utilized the easily obtainable methioninase (METase) as a more straightforward means of creating the MR environment. One study demonstrated [52] the ability of METase to restore sensitivity to Cisplatin in drug-resistant gastric cancer cells. The specific mechanism involves the inhibition of nuclear factor-κB (NF-κB) activity in gastric cancer cells after treatment with METase, resulting in the upregulation of tumor-necrosis-factor-associated apoptosis-inducing ligand (TRAIL) expression and the subsequent downregulation of P-gp. Furthermore, the inactivation of NF-κB results in the downregulation of miR-21 expression, further enhancing the sensitivity of gastric cancer cells to Cisplatin [139]. NF-κB, a transcription factor, can activate molecules such as anti-apoptotic proteins, thus diminishing apoptosis in gastric cancer cells [140]. TRAIL can selectively induce cancer cell death without causing harm to normal cells [141] (Figure 5).

Furthermore, growing clinical evidence suggests MR is a promising new cancer treatment modality [64]. A late phase II randomized multicenter trial involving 138 individuals with advanced gastric cancer showed that treatment with MR total parenteral nutrition for 14 days along with 5-fluorouracil and mito-mycin C resulted in a higher therapeutic response (26.3%) compared to the control group, which was given conventional Met-containing total parenteral nutrition with 5-fluorouracil and mitomycin C (8.1%). The difference was found to be statistically significant [27]. The safety of MR therapy has also been demonstrated. The consecutive brief period of a methionine-restricted (MR) diet combined with chemotherapy did not affect the patients’ nutritional statuses, evidenced by stable body weights and consistently constant plasma albumin and prealbumin levels. The combination also displayed acceptable toxicity, mainly affecting hematological parameters, particularly the platelet count [142]. In addition, a recent case report [143] showed that a 62-year-old female patient with breast cancer experienced a recurrence of metastasis in the axillary lymph nodes after four years of treatment. Due to the localized nature of the metastasis in the axillary lymph nodes, the treatment plan involved administering neoadjuvant chemotherapy comprising 3 months of doxorubicin and cyclophosphamide, followed by 3 months of docetaxel. Additionally, during the 6 months of chemotherapy, the patient underwent MR therapy, involving a low methionine diet and oral rMETase supplementation. The patient did not experience rMETase-related side effects throughout this period, aside from mild adverse effects resulting from chemotherapy.

The evidence presented supports the crucial role of Met in chemotherapy for gastric cancer. Furthermore, MR can enhance the sensitivity of gastric cancer cells to various chemotherapeutic agents by regulating the cell cycle. In addition, MR can enhance the efficacy of chemotherapy through multiple mechanisms. These findings suggest that MR can improve the chemosensitivity and effectiveness of gastric cancer cells without being affected by the heterogeneity of the disease.

## 4. Conclusions and Prospects

The prognosis of advanced gastric cancer remains extremely poor, primarily due to the heterogeneity of gastric cancer cells and the monotony of gastric cancer treatment. The intra-tumoral heterogeneity and inter-tumoral heterogeneity of gastric cancer cells result in suboptimal therapeutic responses to many molecular targeted drugs, with some failing in clinical phase II and phase III trials. Even the targeted drugs currently used in treatment are essentially not suitable as first-line therapies and exhibit unstable efficacy. Therefore, molecular targeted therapy may not be the optimal choice for treating gastric cancer. Met is an essential amino acid for gastric cancer cells and plays a crucial role in the initiation and progression of gastric cancer. Additionally, the methionine cycle, centered around Met, also plays a significant role in gastric cancer. Consequently, limiting the acquisition of Met by gastric cancer cells significantly impacts the proliferation and invasion of gastric cancer. As discussed earlier, creating an MR environment can influence gastric cancer in multiple ways, independent of the heterogeneity of gastric cancer cells.

Based on the recent literature on Met and gastric cancer, we have found that Met’s influence on gastric cancer cells spans the entire process of gastric cancer development. The existing research suggests that, in the initiation stage, the presence of Met favors the survival of Hp and its impact on surrounding normal tissues. However, creating an MR environment through dietary interventions or enzyme addition makes Hp more susceptible to oxidative-stress-induced damage and impairs its survival, as well as mitigates the inflammation caused by Hp, thereby delaying the onset of gastric cancer. Regarding programmed cell death in gastric cancer cells, an MR environment can facilitate various types of programmed cell death, including autophagy, apoptosis, and ferroptosis. In terms of the immune response in gastric cancer, the role of Met is crucial. MR can inhibit the infiltration of monocytes/macrophages, relieve the competitive suppression of cancer cells on T cells, and promote T cell proliferation and differentiation, thus facilitating the T cell-mediated killing of gastric cancer cells. For GCSCs, MR can inhibit their proliferation through multiple mechanisms, thereby suppressing the growth and invasion of gastric cancer. During the chemotherapy process for gastric cancer, MR can enhance the sensitivity of gastric cancer cells to many chemotherapeutic drugs and enhance the efficacy of chemotherapy drugs through various mechanisms. Of course, the clinical application of MR therapy needs to be explored in more clinical trials, but we consider MR to be a vital component of gastric cancer treatment that bypasses the influence of gastric cancer heterogeneity, though further exploration into the underlying mechanisms is warranted. The metabolic therapy represented by MR has the potential to become a new research direction for gastric cancer.

## Figures and Tables

**Figure 1 biomolecules-14-00161-f001:**
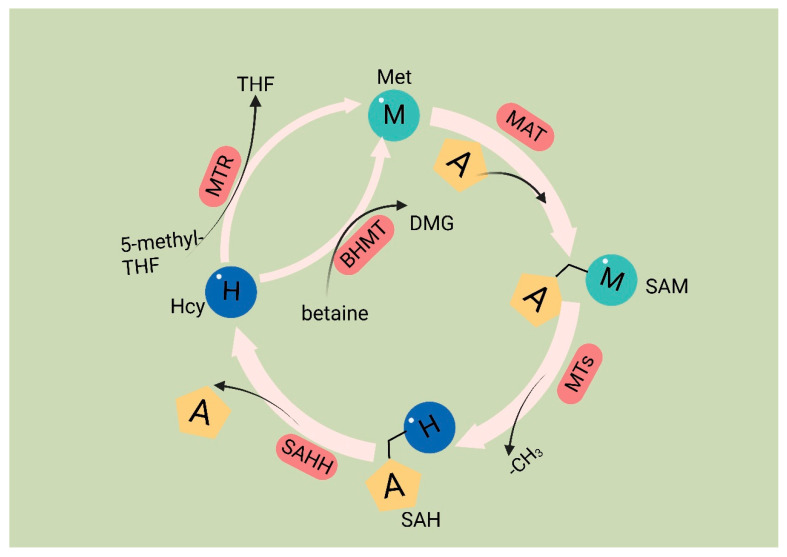
Methionine cycle. The first step of the methionine cycle is the conversion of methionine into S-adenosylmethionine (SAM) by methionine adenosyltransferase (MAT). SAM is converted into S-adenosyl homocysteine (SAH) after donating a methyl group for methylation reactions. This step is mediated by methyltransferases (MTs). SAH is then hydrolyzed by S-adenosyl-L-homocysteine hydrolase (SAHH) to generate homocysteine. Finally, homocysteine receives a methyl group from the folate cycle or betaine to become methionine; these reactions are mediated by 5-methyltetrahydrofolate: homocysteine methyltransferase (MTR) and betaine-homocysteine methyltransferase (BHMT), respectively. The methionine cycle is interconnected with three important metabolic pathways by providing substrates. These pathways include the folate cycle, the transsulfuration pathway, the methionine salvage pathway, and polyamine synthesis, all of which support important cellular functions. Met: methionine; SAM: S-adenosyl methionine; MAT: Met adenosyltransferase; SAH: S-adenosylhomocysteine; MTs: methyltransferases; Hcy: homocysteine; DMG: dimethylglycine; BHMT: betaine-homocysteine methyltransferase; 5-methyl-THF: 5-methyltetrahydrofolate; THF: tetrahydrofolate; MTR: methyltransferase.

**Figure 2 biomolecules-14-00161-f002:**
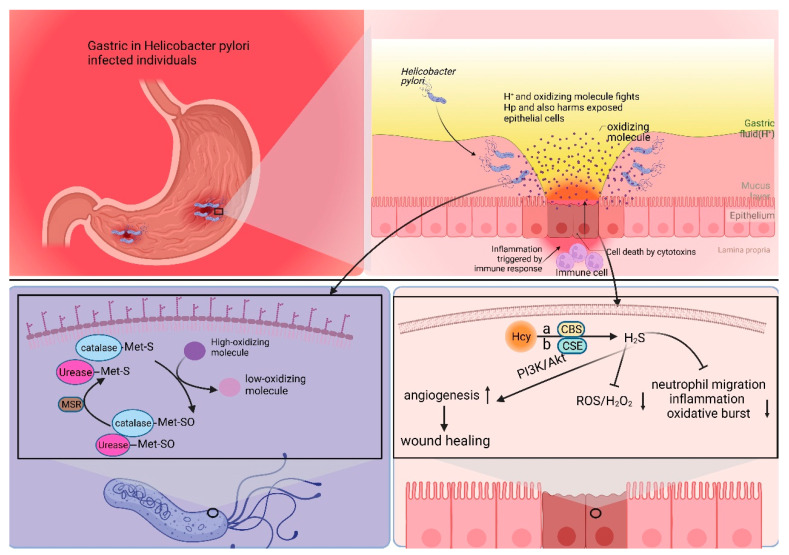
Met plays a pivotal role in the survival and functional repertoire of Hp. Hp infection destroys the mucus layer of the stomach, leading to chronic inflammation within the stomach. During this period, Hp is damaged by both the acidic environment within the stomach and oxidizing factors secreted by inflammatory cells. Hp is able to resist the acidic environment through urease. And it utilizes Met to counteract the antioxidant factors. The S (Met-S) of the Met residue of urease and catalase reacts with oxidizing factors to convert high oxidizing factors to gastric low oxidizing factors, converts itself to Met-SO, and then converts back to Met-S when catalyzed by methionine sulfoxide reductase. Hcy is converted to H2S via three pathways: cystathionine β-synthase (CBS), cystathionine γ-lyase (CSE), and 3-mercaptopyruvate sulfotransferase (3MST). H_2_S is a reducing agent that scavenges oxidizing molecules, including ROS and hydrogen peroxide (H_2_O_2_). Additionally, H_2_S enhances wound healing by facilitating angiogenesis through the phosphatidylinositol 3-kinase (PI3K)/Akt signaling pathway. Furthermore, H_2_S restricts neutrophil migration, inflammation, and oxidative burst.

**Figure 3 biomolecules-14-00161-f003:**
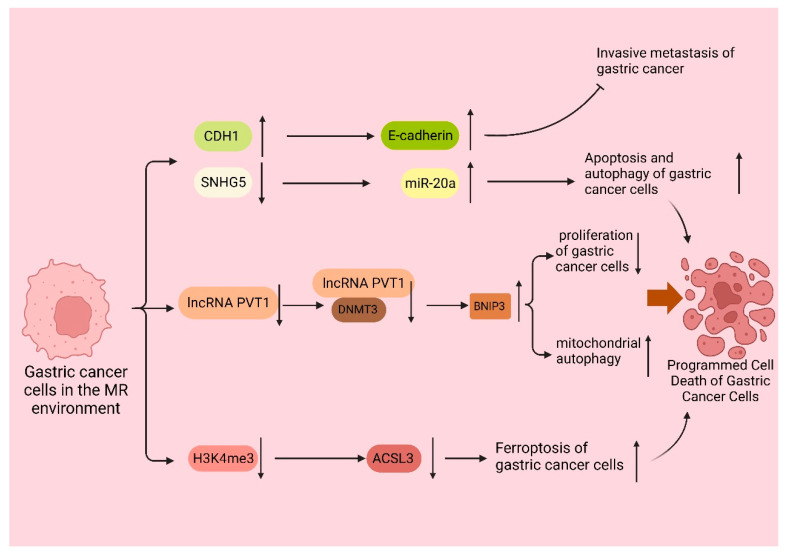
MR promotes the programmed cell death of gastric cancer cells. In the MR environment, the CDH1 promoter was completely demethylated, upregulating the expression of CDH1 and thus enhancing the expression of E-cadherin and inhibiting the invasive metastasis of gastric cancer. The downregulation of SNHG5 expression and the increase in miR-20a expression promoted the autophagy and apoptosis of gastric cancer cells; the expression of long non-coding RNA (lncRNA) PVT1 was also significantly downregulated, and the interaction between lncRNA PVT1 and DNMT3 was weakened. The expression of long non-coding RNA (lncRNA) PVT1 was also significantly downregulated, and the interaction between lncRNA PVT1, and DNMT3 was weakened, leading to the decrease in the DNA methylation level of the BNIP1 promoter and the upregulation of the level of BNIP3, which suppressed the proliferation of gastric cancer cells and activated the autophagy of mitochondria, leading to the apoptosis of gastric cancer cells.

**Figure 4 biomolecules-14-00161-f004:**
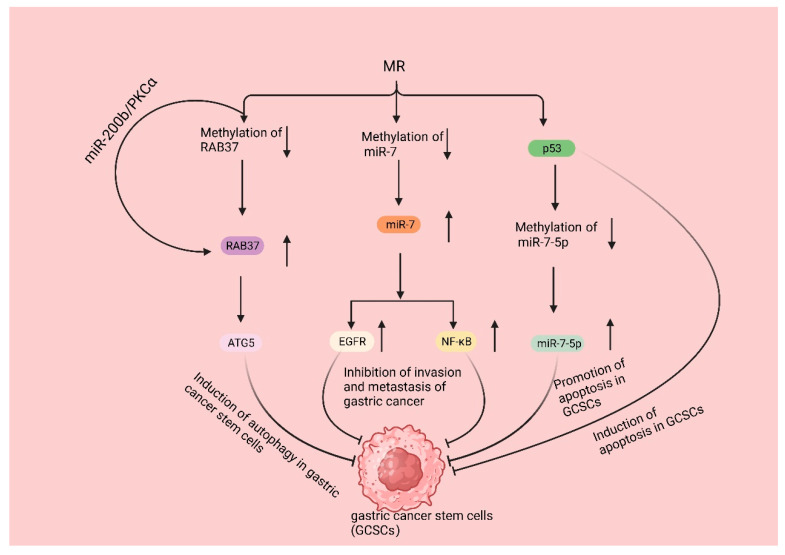
MR suppresses gastric cancer development by inhibiting GSCSs. MR decreases the methylation of RAB37, resulting in the upregulation of RAB37 expression. Additionally, regulating the miR-200b/PKCα axis indirectly increased the activity of RAB37; then, binding it to autophagy-related gene 5 (ATG5) induced the autophagy of GCSCs. MR decreases the methylation of miR-7, resulting in the upregulation of miR7 expression, leading to the inhibition of epidermal growth factor receptor (EGFR) and nuclear factor-κB (NF-κB) signaling pathways, eventually inhibiting gastric cancer invasion and metastasis. MR activates p53 signaling in GCSCs, selectively triggers apoptosis, and causes a decrease in the DNA methylation level of the miR-7-5p promoter region, thereby further promoting apoptosis in GCSCs.

**Figure 5 biomolecules-14-00161-f005:**
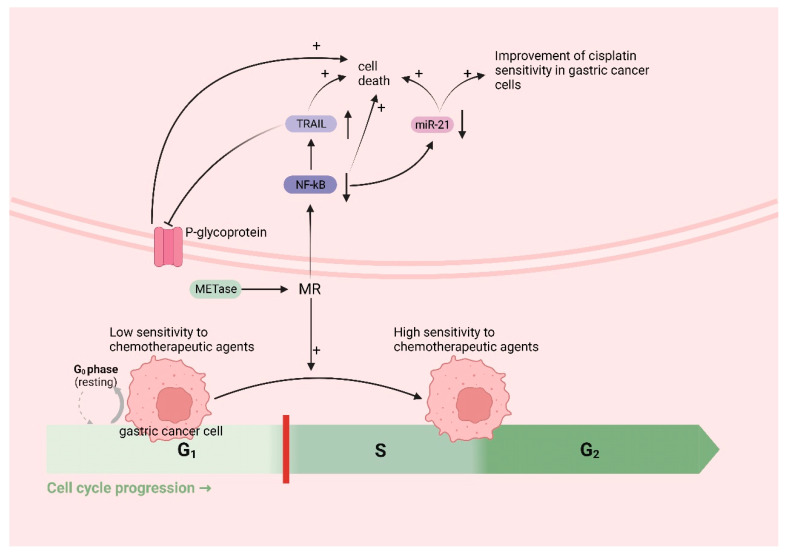
MR improves chemotherapy efficacy in gastric cancer: MR induces the entry of cancer cells, which are trapped in the G0/G1 phase (low sensitivity to chemotherapeutic agents), into the S/G2 phase (high sensitivity to chemotherapeutic agents). MR downregulates nuclear factor-κB (NF-κB), which leads to the upregulation of tumor-necrosis-factor-related apoptosis-inducing ligand (TRAIL) expression, followed by the downregulation of P-gp and the death of cancer cells; the downregulation of miR-21 expression, which further enhances the sensitivity of gastric cancer cells to cisplatin and induces the death of cancer cells; and the downregulation of NF-κB, inducing the death of cancer cells.

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
