# Peer review of "The Role of Methionine Restriction in Gastric Cancer: A Summary of Mechanisms and a Discussion on Tumor Heterogeneity"

_biomolecules, 2024, doi:10.3390/biom14020161_

Round 1
Reviewer 1 Report
Comments and Suggestions for Authors
1. This is a possibly interesting review. Focus more on Methionine, related enzymes like Methinase and Met adenylyltransferase. Too much discussion on Helico and cell cycle.
2. 1. The statement on line 44-49 is strange for me. Did ref 15 and 16 really indicate these one? Gastric cancer incidence is decreasing in many countries and increase in younger population due to bad diet habit, I assume, not valid so far.
3. It would be fair to cite DOI: 10.1134/S0006297923070076.
4. What are MR? (Line 226) and Hcy (Line 225)?
5. Check the MAT2A: some database prefer "methionine adenosytransferase 2A". I am not certain. "sulfate adenylyltransferase" appears in the list. Consult to chemists.
Reviewer 2 Report
Comments and Suggestions for Authors
This is a solid piece of work, and as far as I can tell, thoroughly researched and spiked with a very significant number of relevant references that are mostly up to date. Especially the 1st section on Helicobacter is very detailed and informative.
The next section on apoptosis and variants of cell killing like ferroptosis is entirely or mainly based on in vitro experimental data. This is as such not a bad thing and shows the authors have researched the literature well. Nevertheless, it appears somewhat remote from the clinical situation and any clinically relevant mechanisms. I think the authors should try to make this link, for at least a few examples, as it would definitely strengthen the manuscript by relevance. Are any of the mentioned genes like ACSL3, BNIP3, miRNA 20a, CDH1 of any relevance as markers of patient outcome, disease-free or total survival, or patient cohort classification?
The same applies for some of the genes mentioned in the following chapter(s), such as MiR-7, miR-7-5p, ATG5, RAB37, etc... its ice tohave these candidates popping up in in vitro studies, but if ANY link between in vitro experiments and clinical issues can be made, this would be a very valid issue to be pointed out and of great (er) interest to the readers than "only" in vitro studies. Clinicans may be searching for clues in your review, just make their lives easier and provide them with some...
To me, the most interesting chapter relates to the chemotherapy targeting MET-related mechanisms and pathways.
The authors point towards interesting mechanisms, like the TRAIL-related signalling, but they could also highlight a few other things that become relatively obvious from even a shrt literature research, and which relate to experimental therapies directly targeting MET-related metabolism. Currently, the authors point to the link between OTHER therapies with repercussions that feed back into MET pathway. Not all of this is very convincing, since its again mostly in vitro stuff and choing a bunch of other cell lines may result in very different data... as usual.
The jump from apoptosis to cancer immunology couldnt be more extreme! There is apparently no link between these chapters, and its a bit unclear if and how they would be related. That should be more clearly pointed out.
I think for many readers, experimental threrapies that are targeting MET metabolism this way or another would be really interesting, and there are a few that pop up when you do a little literature research. Some are briefly mentioned but ost arent really elaborated. And this may be important in the future, some of these approaches might be more or less successful, especially when combined with other therapies. Examples of this might be:
Dietary methionine restriction: Reducing dietary intake of methionine can slow down the growth of cancer cells while sparing normal cells. This approach, known as methionine restriction or methionine depletion therapy, is being studied in preclinical models and clinical trials.
Methionine Adenosyltransferase (MAT) Inhibitors: MAT is an enzyme that plays a crucial role in the conversion of methionine to S-adenosylmethionine (SAM), which is a methyl donor for DNA and protein methylation reactions. Inhibiting MAT can disrupt these processes in cancer cells, potentially slowing down their growth. Some compounds that target MAT are under investigation.
Methionine Salvage Pathway Inhibitors: The methionine salvage pathway allows cells to recycle methionine from methylthioadenosine (MTA). Inhibiting this pathway can lead to methionine deprivation in cancer cells. Drugs targeting enzymes in this pathway are being explored as potential therapies.
Methionine Analogues: Developing methionine analogs that can be incorporated into proteins during synthesis but disrupt protein function can selectively target cancer cells. This approach aims to "trick" cancer cells into incorporating these analogs into their proteins, leading to dysfunction and cell death.
Targeting Methionine Transporters: Methionine transporters play a crucial role in supplying cancer cells with methionine from the bloodstream. Inhibiting these transporters can limit methionine uptake and disrupt cancer cell metabolism.
Combination Therapies: Combining methionine metabolism-targeting therapies with other conventional cancer treatments, such as chemotherapy or radiation therapy, may enhance their effectiveness. For example, methionine restriction combined with chemotherapy has shown promise in preclinical studies.
Round 2
Reviewer 1 Report
Comments and Suggestions for Authors
The revisions are acceptable.